# Resilient or Vulnerable? Effects of the COVID-19 Crisis on the Mental Health of Refugees in Germany

**DOI:** 10.3390/ijerph19127409

**Published:** 2022-06-16

**Authors:** Laura Goßner, Yuliya Kosyakova, Marie-Christine Laible

**Affiliations:** 1Institute for Employment Research (IAB) of the Federal Employment Agency (BA), 90478 Nuremberg, Germany; yuliya.kosyakova@iab.de; 2Chair of Sociology, Area Societal Stratification, Otto-Friedrich University of Bamberg, 96052 Bamberg, Germany; 3Research Data Centre of the Federal Office for Migration and Refugees (BAMF), 90461 Nuremberg, Germany; marie-christine.laible@bamf.bund.de

**Keywords:** mental health, refugees, COVID-19, lockdown, Germany, life satisfaction, resilience, crisis

## Abstract

Even though the COVID-19 pandemic had consequences for the whole society, like during most crises, some population groups tended to be disproportionally affected. We rely on the most recent data from the IAB-BAMF-SOEP Survey of Refugees to explore the resilience or vulnerability of refugees in the face of the pandemic. As the 2020 wave of the survey was in the field when the second nationwide lockdown started in December, we are able to apply a regression discontinuity design to analyze how refugees in Germany are coping with these measures. Our results reveal a negative effect of the lockdown on refugees’ life satisfaction. Male refugees and those with a weaker support system face stronger negative outcomes than their counterparts. Since mental health is an important prerequisite for all forms of integration, understanding the related psychological needs in times of crisis can be highly important for policymakers and other stakeholders.

## 1. Introduction

As a global crisis, the COVID-19 pandemic has increased psychological distress and posed challenges to the mental health of many individuals [1,2]. The risk of infection and possibly severe physical consequences induced worries about one’s own health and the health of others. Social isolation due to necessary restrictions of social contacts as well as uncertainties regarding future policy measures likely placed further burdens on mental health (for an overview, see [3]). Even though the pandemic had substantial consequences for the whole society, like during most crises, some population groups tended to be disproportionally affected. In particular, personal circumstances and correspondingly the degree to which individuals are impacted by the pandemic vary. For instance, various studies showed that parents whose children could not continue visiting care facilities had to take on a higher burden than, e.g., non-parents [4,5]. Yet, even under similar circumstances, some individuals might cope more easily with mental distress, while others are more adversely influenced by it. This resilience or vulnerability in the face of the pandemic is what we focus on in our study. More specifically, we examine the effect of Germany’s second nationwide lockdown on refugees’ life satisfaction. By doing so, we first elaborate on whether or not refugees represent a resilient or vulnerable group in the face of the pandemic. Second, we explore factors that enhance the resilience of refugees.

In the literature, resilience is generally understood as an effective response to stressful events [6]. Nevertheless, the concept of resilience has been applied in different ways by previous researchers. Following a systematic review of the adult mental health literature [7], resilience has been understood as either a process or as a personal resource of an individual. Studies that conceptualize resilience as a process examine the ability to withstand [8], recover [9], or even grow [10] from disturbing events. In studies where resilience is conceptualized as a personal resource, the research focuses on underlying personal or social factors [11] that enhance resilience. Both concepts can also overlap or be used simultaneously [6]. Against the background of these different understandings of the term resilience, it is of particular importance to clarify the concept of resilience we use throughout the paper. In this study, we contribute to both strands of literature. Following a widely applied definition by Bonanno [12], we define resilience as the ability to keep a stable level of mental functioning, even when experiencing disturbing events. The analyses of our paper are divided into two steps. First, we analyze the effect of the lockdown on mental health. For this purpose, we align ourselves with the strand of the literature that conceptualizes resilience as a process [7], more specifically, as the ability to withstand disturbing events. Regarding the interpretation of our results, individuals are therefore considered to be resilient if they retain stable levels of mental health, measured by life satisfaction, in the face of the pandemic. At the same time, we follow the previous literature [13] and conceptualize resilience and vulnerability as opposite outcomes. Hence, a decrease in life satisfaction in the face of the pandemic points to the individual’s vulnerability. Second, we explore the heterogeneity of the lockdown’s effect on mental health and examine which external and internal factors enhance resilience. We hereby also contribute to the alternative strand of literature where resilience is considered as a resource. We consider factors or characteristics that are related to a (stronger) decrease in life satisfaction as resilience-enhancing. Factors or characteristics that are related to no (or a less severe) decrease in life satisfaction point to the vulnerability of the group.

To examine the individual responses to resilience, previous research analyzed resilience in the aftermath of potentially traumatic events, such as natural disasters [14] or disruptive family events, such as parents’ divorce [15] or the loss of a spouse [16]. We expand this literature by looking at the potentially disruptive event of the second nationwide lockdown in Germany in December 2020. To reduce the number of SARS-CoV-2 infections and minimize the outbreak of the virus during the COVID-19 pandemic, many governments decided to impose lockdowns that restricted the daily lives of people and affected the whole society. While such lockdowns were effective in the prevention of deaths and infections [17], it is equally important to understand related psychological effects.

We focus on the coping strategies of refugees for several reasons. First, previous findings suggest that, alongside migrants in general, refugees face disproportionally high risks for COVID-19, for example due to precarious employment situations, crowded accommodations, or barriers to access the health system [18,19,20]. Second, refugees face not only additional worries for families and friends in the home country [21,22], insecurity about their asylum status and prospects to stay [23,24], or little established support systems [22] but also a lack of host-country language skills [25] and fewer economic opportunities [26]. All those factors can put additional strain on mental health. Third, there is a common criticism that the Western literature tends to pathologize refugees by pointing the focus away from their possible strengths and depicting them solely as traumatized victims [27,28]. As refugees’ migration process is often coupled with trauma and stress before and during flight, they are likely to have undergone self-selection mechanisms and to have developed coping strategies through previous experiences [29,30]. Hence, we consider refugees’ previous experiences by looking at factors that might enhance resilience. In particular, we consider the role of personal characteristics such as an internal locus of control or protective factors such as an elaborate support system.

To examine whether the second nationwide lockdown in Germany in December 2020 affected refugees’ life satisfaction, we rely on the most recent survey data from the IAB-BAMF-SOEP Survey of Refugees and implement a regression discontinuity (RD) design. The timing of data collection in our study provides a unique opportunity for causal inference because refugees were interviewed before as well as after the introduction of the lockdown. We further explore the heterogeneity of the lockdown’s effect to uncover specific internal and external protective factors in favor of resilience. Our results reveal that the introduction of the lockdown in the calendar week 51 of 2020 significantly decreased refugees’ life satisfaction. Moreover, male refugees and refugees with a well-established support system—indicated by the number of people to rely on—seem to be less affected than those without.

By examining one of the consequences of the COVID-19 pandemic, namely the introduction of a nationwide lockdown, on the mental health of refugees, this paper contributes to a better understanding of the consequences of crises on individual well-being. Since crises can put a significant strain on mental health, understanding the vulnerability or resilience of specific groups (such as refugees) and the factors that enhance resilience are of utmost importance. From the policy perspective, the identification of vulnerable groups is extremely relevant to providing targeted and timely assistance to those in need and assisting in mitigating the consequences of the crisis. With our study, we provide the necessary insights to assess whether refugees should be considered a vulnerable group to which additional resources should be allocated in times of crisis. While previous research suggests that refugees face high psychological burdens during the COVID-19 pandemic [20,31], we are not aware of any studies that were able to uncover causal effects.

The remainder of this paper is organized as follows: Section 2.1 presents the data used for the empirical analyses, Section 2.2 gives details on the outcome variable life satisfaction, Section 2.3 explains the events happening in Germany surrounding the second nationwide lockdown, Section 2.4 presents details on our empirical method, and Section 2.5 gives background information on underlying factors we will use for the heterogeneity analysis. We present our results and robustness checks in Section 3, a discussion of results in Section 4, and the conclusion drawn from our study in Section 5.

## 2. Materials and Methods

### 2.1. IAB-BAMF-SOEP Survey of Refugees

Our empirical analyses are based on the IAB-BAMF-SOEP Survey of Refugees [32]. This large-scale longitudinal survey of refugees and their household members in Germany was launched in 2016 and is conducted annually. The sample was drawn from the Central Register of Foreigners and is representative of adult refugees who arrived in Germany between 1 January 2013 and 31 December 2016 (irrespective of their current legal status). This includes individuals with pending asylum decisions (asylum-seekers), individuals with positive asylum decisions that can be made on different legal grounds (political asylum according to Art. 16a of the German Constitution, refugee status as defined by the 1951 Refugee Convention, subsidiary protection, and national ban on deportation), or individuals who have received negative asylum decisions but whose deportation has been temporarily suspended. For further details, refer to the work of Kosyakova and Brücker [33]. By using appropriate sample weights, the data allow us to make representative inferences for this refugee population in Germany and their household members [34].

By wave 5 in 2020, a total of around 22,500 observations had been collected. Interviews were conducted face-to-face with computer assistance (CAPI) and were supported by translators if needed. Questionnaires were available in seven languages (Arabic, English, Farsi/Dari, German, Kurmanji, Pashtu, and Urdu) and with auditory instruments for illiterate respondents. Because of the pandemic, interviewers and respondents could also choose to conduct the interview via telephone (CAPI-TEL) in the 2020 wave. The fieldwork traditionally started in the second half of the corresponding survey year: wave 1, fieldwork period June–December 2016; wave 2, fieldwork period June 2017–March 2018; wave 3, fieldwork period September 2018–February 2018; wave 4, fieldwork period August 2019–January 2020; and wave 5, fieldwork period August 2020–February 2021. The field periods’ timing enables us to implement our regression discontinuity design because wave 5 was in the field when the second nationwide lockdown started in December (see Section 2.3).

Our empirical analyses are concentrated on wave 5 (i.e., survey year 2020). As we consider pre-pandemic baseline characteristics, our sample includes respondents who participated in both survey waves 4 (i.e., survey year 2019) and 5 (i.e., survey year 2020). We exclude respondents who arrived in Germany before 2013 or who did not come to Germany as asylum seekers, as well as respondents with implausible information on their asylum procedure. After these exclusions, we obtain 2540 person observations, i.e., 5080 person-year observations.

### 2.2. Variable of Interest: Life Satisfaction

To address refugees’ mental health, we rely on life satisfaction. Life satisfaction is understood as one of the major components of subjective well-being and has been used widely in psychological research as an indicator of mental health [24,35,36]. Previous studies have also used this item to assess resilience [37,38,39]. Respondents are asked how satisfied they currently are with their life in general. Answers are given on an 11-point scale that ranges from 0 (completely dissatisfied) to 10 (completely satisfied), such that higher values are interpreted as higher levels of mental health. By using life satisfaction as a measure of well-being, we align ourselves with studies that take a hedonic approach and understand well-being as subjective happiness and pleasure. On the opposite strand of literature—where researchers take a eudaimonic approach—well-being is understood to come from self-fulfillment and meaning of life [40]. While we agree that both strands—hedonia as well as eudaimonia—can contribute to psychological well-being in their own way [41], we focus on life satisfaction because this approach allows us to embed our results in the prior literature on resilience.

Figure 1 shows how the life satisfaction of respondents in the IAB-BAMF-SOEP Survey of Refugees has developed over time. The mean of life satisfaction is calculated per interview week of each survey year (weeks with less than 10 observations are excluded from the graphic). Panel (a) shows the life satisfaction of refugees in the sample used for our empirical analysis. The figure shows an increasing trend in life satisfaction. A similar trend also arises if we consider the full data from 2016 to 2020 in panel (b).

At a first glance, we observe no drop in the level of life satisfaction from 2019 to 2020 that could be attributed to the pandemic. However, we will take a closer look at the introduction of the second nationwide lockdown in Germany in December 2020 to obtain a clearer picture of the lockdown’s effects.

### 2.3. Second Nationwide Lockdown in Germany

In our empirical analyses, we focus on the introduction of the second nationwide lockdown in Germany. As in many other countries, the German government responded to increasing numbers of infections with measures such as restrictions on social contacts, closures of schools, and closures of non-essential shops. The first lockdown was introduced on the 22nd of March 2020 and lasted several months [42]. During the summer, infection rates decreased substantially [43] such that it was possible to come back to a relatively normal day-to-day life with some preventative measures such as keeping distance, observing hygiene rules, and wearing everyday masks. By the end of October, the situation had worsened again. In response to the exponentially rising COVID-19 incidence rates, the government first introduced the so-called “*Lockdown Light*” on November the 2nd [44]. The included measures only showed partial success, and by the middle of December, infection rates increased exponentially again [43]. Against this background, the Chancellor and the Minister Presidents of the federal states agreed on additional and more restrictive measures that marked the beginning of the second nationwide lockdown in Germany [45]. Among others, the following new measures were announced on December 13th: private gatherings were limited to one’s own household and one other household, but in any case, to a maximum of five persons; non-essential retail shops and restaurants were closed; personal care services such as hairdressers were closed; schools and daycare centers were closed; and employers were urged to close operating sites through home-office regulations or company vacations. Measures came into force on the 16th of December and were gradually loosened as of March 2021 [46].

The previous literature stresses that lockdown measures negatively affected the psychological well-being of individuals in many countries (see [3] for an overview). Correspondingly, in our empirical strategy, we assume that this lockdown affected the lives of large (if not all) parts of Germany’s society (see [31,47,48] for empirical support). Nevertheless, we cannot observe who complied with the rules of the lockdown or who did not. We also cannot rule out if some individuals had already taken restrictive measures on their own such that the introduction of the lockdown was not of any concern to them. For an impression of how strongly the introduction of the lockdown was perceived in German society, we take a look at Google Trends data for the word “Lockdown” visualized in Figure 2. As Figure 2 illustrates, the search frequencies on the term lockdown increased at the end of October 2020, with the second expansion in mid-December 2020. This surge in search frequencies provides a hint that the lockdown measures were strongly noticed in society. Likewise, Figure 2 hints at the fact that the December lockdown instead of the so-called “*Lockdown Light*” in November was more perceived by the German society, as the search volume for the term in the latter period was only half of what it was in December.

### 2.4. Regression Discontinuity Design

To study the effect of the lockdown, the statistically most preferable method would be to conduct an experiment where we randomly assign the treatment (i.e., living under the lockdown) to half of the individuals and compare their outcomes to a control group. However, such an experiment would be neither technically feasible nor ethically acceptable to conduct. Instead, the timing of the fifth wave of the IAB-BAMF-SOEP Survey of Refugees provides a natural experiment that allows us to implement a regression discontinuity (RD) design. Thus, we can study the causal effect of the lockdown on life satisfaction by comparing individuals interviewed before and after the start of the lockdown. Previous research implemented the RD design in a similar way; for example, Van Hauwaert and Huber [49] estimated the effects of the Paris terror attacks in November 2015 by analyzing survey data that were in the field at the time of the events. Figure 3 depicts the timing of interviews during the fifth wave for individuals who participated in 2019 as well.

The interview period in 2020 extended from the 24th of August 2020 until the 15th of February 2021, while the lockdown was introduced in calendar week 51. We use this week as the cutoff to define the treatment assignment. We decided in favor of interview weeks instead of dates to specify the cutoff because the start of the lockdown was announced three days ahead. Specifying the 16th of December as our cutoff could bias our results, as many people would likely have already anticipated the treatment. Since we are analyzing the psychological effects of the lockdown—which also stem from uncertainty regarding measures and not purely from their implementation—we hypothesize that the announcement itself could already have led to psychological distress. Correspondingly, interview week 51 defines our treatment assignment. With this specification, 74.8% of individuals belong to our control group and were interviewed before the cutoff.

One of the main assumptions of the RD design is that individuals have no influence on their treatment assignment. Hence, we assume the interview week to be exogenous. Figure 3 provides support for our assumption: the number of interviews does not immediately jump up or down at the cutoff. We also test our assumption with a density test using a manipulation testing procedure with local polynomial density estimators [50,51]. The null hypothesis that the density is the same above and below the cutoff (i.e., no manipulation) cannot be rejected (test-statistic T = −0.3275, *p*-value *p* = 0.7433), such that we find no sign of manipulation. To further assure that people were assigned to the treatment or control group randomly, we next conduct a balance check. For this, we first conduct a *t*-test to test for significant differences in pre-treatment characteristics—variables that are either invariant or were measured in the 2019 wave—among treatment and control groups. Results are presented in column (3) of Table 1. Significant differences exist for several variables. Nevertheless, following Imbens and Lemieux [52], pre-treatment characteristics can vary on average but should not show discontinuities at the threshold. We test this assumption by using each variable as a placebo outcome and find no significant discontinuities at the cutoff. Results are reported in column (4) of Table 1.

When applying an RD design, a decision between the so-called *sharp* or *fuzzy designs* is necessary [52]. In the sharp RD design, the probability of treatment is assumed to jump from 0 to 1 at the cutoff. In the fuzzy RD design, a jump in the treatment probability still occurs but is not necessarily from 0 to 1; i.e., compliance with treatment is allowed to be imperfect. This second assumption holds in our case for two reasons: First, individuals might not follow the rules of the lockdown. Thus, we would observe individuals in the treatment group that were not actually treated. Second, strict rules of (self-)isolation could be voluntarily implemented by individuals or enforced by authorities. Thus, we would observe individuals in the control group that were treated, as they experienced a lockdown of their own. Nonetheless, as the Google Trend data (Figure 2) indicated that the search volume for the word “Lockdown” increased abruptly at the cutoff, we still assume a jump in treatment probability. However, for the given reasons, we do not assume this jump to be from 0 to 1. To fully implement a fuzzy RD design, we would need information on who complied with the treatment. Unfortunately, this information is unavailable; therefore, we implement the sharp RD design but interpret our results as intention-to-treat (ITT) effects [53].

One challenge of the RD design is the necessity to make assumptions about the functional form of the outcome variable around the cutoff. To avoid this difficulty, we implement local polynomial analysis as a non-parametric approach that allows avoiding assumptions about a particular form of life satisfaction by approximating it instead. For the approximation, two regressions are estimated on either side of the cutoff. The treatment effect then reflects the difference between intercepts of the estimations. We follow the standard approach of using a linear model for the choice of the polynomial’s order [54]. Furthermore, we apply mean squared error (MSE) optimal bandwidths. Choosing the bandwidth is subject to a trade-off between bias and variance, where fewer observations lead to an increase in variance and the use of more observations leads to an increase in bias. We choose the bandwidth that minimizes the approximate MSE of the point estimator, dependent on the chosen order of the polynomial (first order). We apply STATA’s user-written program rdrobust for this procedure [55].

Additionally, we test specifications with both triangular and uniform kernel weights. To increase the precision of the point estimator, we optionally include the control variables gender, age at arrival, years of schooling before migration, partnership in 2019, children in household in 2019, language proficiency in 2019, and life satisfaction in 2019 in the estimation. In the RD design, control variables are required to be predetermined and independent of the treatment variable; i.e., they must be continuous at the cutoff [56]. We tested this requirement with the placebo outcome tests presented in Table 1.

### 2.5. Heterogeneity Analysis

In the second step, we conduct a heterogeneity analysis to explore factors that might be related to the resilience or vulnerability of refugees. As mentioned in the introduction, we hereby conceptualize resilience as a resource and study which factors are positively related to it. Note that both pandemic-related stressors and general protective factors might play a role in enhancing resilience. Factors that are, e.g., related to a negative effect and may enhance vulnerability can do so through two different mechanisms: On the one hand, these factors can characterize subgroups that are more affected by the pandemic as such, e.g., because those subgroups face higher risks of infection. On the other hand, such factors can be characteristic of subgroups that are less able to cope with the pandemic. Both mechanisms are of equal importance for any conclusions we might draw. Thus, we do not distinguish between the two: rather, we focus on external and internal protective factors. This focus follows a widely used distinction in resilience research [57].

Among the external protective factors, we first test for differences in the effect by socio-economic variables, namely gender, employment status, job level, and children in the household, as the pandemic likely affected these subgroups differently [3]. Next, we look at two more commonly researched external factors related to support systems improving resilience. Here, we consider differences in the number of persons the respondent indicates sharing personal thoughts with, and differences in partnership status. Additionally, we look at language proficiency as an external protective factor. Both support system and language proficiency can enhance resilience among refugees [28].

The literature also finds positive effects for some internal protective factors on resilience. We test if differences in the outcome exist due to locus of control. The concept of locus of control was first introduced by Julian Rotter [58] and describes the degree to which individuals think they have control over things happening in their lives. Individuals with an internal locus of control believe that they are responsible for their lives even under difficult circumstances. Individuals with an external locus of control attribute events in their lives to pure faith or luck, on which they consequently cannot exert any influence. Previous research shows that an internal locus of control is beneficial to resilience in stressful situations [59,60]. For more details on the definition and measurement of our variables and descriptive statistics, see Appendix A.

For each subgroup, we conduct separate RD estimations. Following the main analysis, we use a local polynomial analysis of order 1 with triangular kernel weights, MSE-optimal bandwidths, and sample weights. To increase the point estimator’s precision, we control for life satisfaction in 2019. To prevent possibly biased effects due to reverse causality, we use the 2019 observation for all factors that are time-variant and are likely to have changed due to the lockdown itself. The remaining variables—gender, locus of control, and children in the household—are time-invariant or not subject to change due to the introduction of lockdown.

## 3. Results

### 3.1. RD Estimation Results

Figure 4 illustrates the estimated effect of the second nationwide lockdown in Germany on refugees’ life satisfaction with an RD plot. The dots represent optimally chosen binned means of life satisfaction, and the solid blue line shows the polynomial approximation for control and treatment observations separately [61].

The plot indicates that life satisfaction decreased when the lockdown was introduced. Nevertheless, this plot does not allow interpretations of the result’s significance and does not confirm the effect. Using the RD estimation, Table 2 presents more formal results of how the lockdown impacted the life satisfaction of refugees in Germany.

The RD estimates (see Table 2) confirm a significant negative effect of the lockdown on life satisfaction. We interpret this effect as an ITT effect; i.e., we capture the causal effect of being assigned to treatment. The effect is robust to specifications with and without covariates as well as specifications with uniform and triangular kernel weights. For the specifications with uniform kernel weights, the estimates are significant at the 1% level; for the specifications with triangular kernel weights, the estimates are significant at the 0.1% level. According to our estimates, the life satisfaction of refugees in Germany decreased on average by between 0.8 (${\hat{ß}}_{RD\;}in\;SD$ = 0.4) and 1.1 (${\hat{ß}}_{RD}\;in\;SD$ = 0.6) points on the scale from 0 to 10 when the lockdown measures were introduced in December 2020. As we conceptualized resilience as the ability to keep stable levels of life satisfaction when experiencing the events of the lockdown, we consequently interpret this negative effect as vulnerability amongst refugees following this event.

### 3.2. Heterogeneity of Results

Next, we examine the effect’s heterogeneity by analyzing underlying external and internal protective factors. The results are presented in Figure 5. RD estimates and robust confidence intervals are presented for each subgroup.

For most factors, we find different effect sizes. For example, male refugees; employed refugees, especially those employed as skilled workers; refugees in partnerships; individuals with fewer trusted people around them; refugees with (very) good language proficiency; and refugees with an external locus of control display stronger negative effects than their respective opposites. However, in most cases, the differences are not statistically significant, except for the factors gender, number of trusted persons, and German proficiency, as indicated by the non-overlapping confidence intervals. We may therefore derive three important conclusions. First, the living situations of refugees differ by gender: lockdown reduced life satisfaction of male refugees significantly more strongly than that of female refugees. Second, refugees with a well-established support system (as indicated by more people around them to trust with their feelings and worries) are less negatively affected by the lockdown than those people without such support. Our analyses thereby confirm that support systems are a protective factor enhancing resilience. Third, refugees with higher proficiency in German are significantly more negatively affected by the lockdown than others. Thus, we cannot constitute language proficiency alone as a protective factor. It is for example likely that refugees with better language skills also belong to groups that faced stronger pandemic stressors due to their employment situation. For instance, previous research suggests that workers employed in the so-called interactive tasks that require oral or writing proficiency (e.g., in consulting, educating or teaching, sales) [62] were not only less likely to benefit from home-office possibilities, but also subject to increased health risks and disproportionally affected by company closures [63]. However, further research is needed to understand this finding in detail.

### 3.3. Robustness Checks

To lend further credibility to our findings, we conduct three robustness checks. First, we test for jumps at interview weeks where no jumps are expected [52]; i.e., we test two weeks before and two after the original cutoff. On both sides of the cutoff, we first choose the week that is as far away from the cutoff and as close to the beginning (week 35) or end (week 07) of the interview period as possible. For this selection, we consider a bandwidth of five weeks such that bandwidths are still allowed to be chosen MSE-optimally. This leaves us with week 40 and week 02 on the respective sides of the cutoff. As week 40 lies at the end of the month and week 02 at the middle of the month, we also aim to cover the respective other period on each side of the cutoff. Therefore, we also add week 47 and week 53 to the test, as each also represents the closest possible option to the original cutoff while still fulfilling our requirements. We use the same specification as in the original estimations. Table 3 presents the result of the RD estimations with these placebo cutoffs. We cannot find a significant effect on life satisfaction for any of the cutoffs, suggesting that the assumption of continuity in the absence of the treatment is appropriate.

As a second robustness check, we implement a different estimation procedure for the RD design and combine it with the difference-in-difference (DiD) method. As one of the RD design’s difficulties lies in the necessary assumptions about the functional form of the outcome variable, we used the solution of local polynomial analysis. However, our data also allow an alternative solution: Due to the panel data structure, we can look at previous years to observe how life satisfaction usually varies during the year by combining the RD design with the DiD approach [64,65]. Hereby, the seasonal trend is factually canceled out. As in this procedure the bandwidth is not chosen MSE-optimally anymore, we consider different specifications of bandwidths, namely of five, four, three, and two weeks around the lockdown. The results presented in Table 4 show significant estimates that are in a comparable range to our previous estimations in the main analysis.

Third, we test if the interview mode changed abruptly with the beginning of the lockdown. One major difficulty for the validity of RD design is that there might be other abrupt changes happening at the time of the treatment which then drive the results. To ensure that no other major changes are responsible for our results, we have to rely on adequate research about the period. According to our investigations, the only change that could also have happened at the time of the lockdown is a change in the interview mode. In the 2020 wave, interviewers and respondents could choose to conduct the interview via telephone (CAPI-TEL). In our baseline RDD sample, 12.09% of observations were collected via this mode. By checking the share of telephone interviews by interview week, we observe that the use of the CAPI-TEL mode started in calendar week 45 (2 November 2020) and increased to a share of 58% in week 06 (8 February 2021). For the interview mode to represent a true treatment, the probability of telephone interviews would need to jump up at the cutoff. Therefore, to rule out discontinuities in the share of telephone interviews around the cutoff in calendar week 51, we conduct an RD estimation with the interview mode as a placebo outcome variable. We find no significant effect in this estimation (see Table 5); i.e., the share of interviews conducted in CAPI-TEL mode has not changed abruptly at our used cutoff. We therefore conclude that our results were not driven by a mode switch.

## 4. Discussion

### 4.1. Interpretation of Results

The COVID-19 pandemic affected the lives of people across the globe. In many European societies, such as Germany, the virus, ensuing anxieties, and subsequent nationwide lockdowns have resulted in a devastating loss of life, physical and mental health impairments, major setbacks in education, and loss of private enterprise. Furthermore, challenges for more vulnerable groups in society, among others women, migrants and refugees, older people, and those in caring roles, grew substantially [3]. Falkenhain et al. as well as Tallarek et al. argued that due to their vulnerable economic position and lack of state support, immigrants and refugees, in particular were disproportionately affected by the pandemic [19,20].

However, the lack of studies relying on a strong causal design undermines drawing empirically well-informed evidence regarding the effects of the COVID-19 pandemic on these vulnerable groups. Additionally, descriptive studies are unable to assign negative outcomes of the pandemic to specific pandemic-related events. So far, the extent of a decrease in well-being stemming, e.g., from job loss, health risks, or preventive measures as such remains unclear. Using the most recent data from the IAB-BAMF-SOEP Survey of Refugees and advanced estimation techniques, we explore how refugees in Germany are coping with lockdowns imposed by the government. As the 2020 wave of the survey was in the field when the second nationwide lockdown started in December, we are able to apply a regression discontinuity design to uncover the causal effect of introducing this lockdown.

Our results reveal a negative effect of the lockdown on refugees’ life satisfaction. In terms of our research question and in line with the definition of resilience applied in this paper (i.e., the ability to keep stable levels of mental well-being in the face of disturbing events), we interpret this negative effect as a sign of vulnerability. With an effect size of around −0.5 standard deviations, the impact is comparable to effects found in previous studies of, e.g., adaption to loss of spouse (−0.4) [35] or symptoms of post-traumatic stress in general among US students (−0.21) [66] on life satisfaction. During the period analyzed, the survey was only in the field for the refugee samples, while fieldwork of other SOEP samples—which also include respondents born in Germany—had already finished. We are therefore unable to directly compare the size of our RDD effect for refugees to other German population groups. For instance, Ahrens and colleagues [48] find in a study among the overall population in the Rhine-Main region that on average, the first lockdown in March 2020 even had a positive effect on mental health, while also pointing out that effects need to be looked at separately for vulnerable groups. With our study, we illustrate that refugees should be considered as one of these vulnerable groups and as such require resources assisting mental health issues.

The lockdown effect also implies some heterogeneity by specific subgroups of refugees. While we would interpret factors or characteristics that are related to a less severe decrease, or no decrease, in life satisfaction as resilience-enhancing, subgroups with more severe drops in life satisfaction point to the vulnerability of these subgroups. For instance, we uncover that specifically male refugees are impacted more strongly by the pandemic. As mentioned, we are unable to disentangle the mechanisms behind any heterogeneity, i.e., whether they stem from differences in pandemic stressors or coping abilities. Other studies examining resilience have also pointed to gender-specific effects of the pandemic. For instance, Baguri and colleagues [67] show that female teachers in Malaysia were more resilient regarding dispositional hope than male teachers during the pandemic. Similar patterns might explain our finding. Another explanation might be that because of their higher pre-pandemic employment rate, male refugees were more concerned about being at higher risk of losing their jobs, particularly given refugees’ rather precarious employment situations. More research on gender differences regarding resilience among refugees would need to be done to fully understand the underlying mechanisms. In general, this finding gives incentive to take on gender-differentiated views upon empirical research or implementing policy measures.

The introduction of the lockdown contributes to high levels of uncertainty, as it is unclear how long measures will be in place. In addition, such severe policy measures clearly demonstrate the gravity of the situation, such that fears for one’s health or that of others can rise. We hypothesized that a well-established support system can help to deal with this situation. Accordingly, we find significant differences in the estimated effect for the number of trusted individuals a respondent is able to rely on. Respondents with fewer trusted contacts face stronger impacts of the lockdown on life satisfaction. This finding gives additional support to the literature which considers support systems as an important resilience-enhancing factor [28].

### 4.2. Policy Implications

At least four policy implications follow from our results. First, our results stress the importance of considering mental health issues during crises. Far-reaching (physical) measures required to contain a crisis can impose heavy mental burdens that influence well-being. As psychological well-being is essential for various aspects of life—and maybe even more so in times of crisis—it should not be neglected in policy planning.

Second, we show that refugees present a subgroup that is, at least to some extent, characterized by vulnerability. This vulnerability should be considered when implementing emergency response measures in times of crisis. Therefore, evaluating the different living realities of certain societal groups and providing additional support to those in need is crucial for coping with the consequences of crises.

Third, not all refugees react to crises equally. Thus, we recommend differentiating even amongst the group of refugees when analyzing supportive needs and adopting adequate measures. The heterogeneity of our results for certain subgroups stresses the importance of acknowledging the different challenges some refugees face while others might not. The pandemic has brought out inequalities in the German society, for example between men and women, and these differences might be transferred to refugees as well, making subgroup differentiation complex but important for addressing all issues an individual might face.

Fourth, our results show that the positive effect a support system can have on the mental health of refugees in times of crisis should not be underestimated. This point is also strengthened by previous literature that likewise found support systems to be a resilience-enhancing factor among refugees [28]. Accordingly, independent of crisis response measures, establishing stable, trusted, and reliable contacts and networks should play an important role in the integration of refugees. In addition, contact with family and friends in the home country can be of equal importance for this support system, indicating the necessity of good communication infrastructure.

### 4.3. Limitations

Despite having found a negative effect of the lockdown on life satisfaction, our analysis also has limitations that should not be ignored. First, the RD estimator—by definition—uncovers local effects [52]. Individuals close to the cutoff are compared to each other. For this reason, external validity in RD designs is low, allowing learning about a possible effect close to the cutoff only. In our setting, this means that we cannot make any assumptions about the long-term effects of the lockdown or about how fast refugees might have recovered from this immediate negative effect. Second, like many other studies [7], we define resilience as the ability to withstand disturbing events. We can consequently interpret the significant negative effect on life satisfaction as a sign of vulnerability. However, our results are not comparable to those from studies that define resilience as the ability to recover from stressful events. Under this alternative definition, an RD design would not be the appropriate method for analyzing resilience.

Third, note that our results are specific to Germany. The measures taken by governments during the COVID-19 pandemic differed substantially by country. Additionally, the overall situation for refugees is subject to major differences in other countries. Therefore, we cannot state that lockdowns in other countries would lead to the same effect, to an effect of the same direction, or even to an effect at all. Due to the various numbers of measures included in the lockdown, we also cannot draw any conclusions regarding whether one specific measure drove the results or whether the overall effect comes from the accumulation of all measures combined.

Finally, we want to stress the limitations that come with the use of quantitative methods and with the use of life satisfaction as an indicator of mental health. Standardized surveys might be less effective in capturing the complexity of individual experiences. Quantitative methods therefore also have their limits when it comes to investigating underlying mechanisms. Accordingly, we encourage further investigations using qualitative or mixed-method approaches to shed even more light on aspects of mental health during the pandemic and related lockdown measures. Additionally, though used commonly, life satisfaction is only one of many indicators of mental health. Hence, we encourage future research to consider further measures of mental health such as the Depression, Anxiety, and Stress Scale (DASS)-21 [68] or other commonly used measurement scales.

## 5. Conclusions

By implementing a regression discontinuity design, our study provides empirical evidence on a causal negative effect of Germany’s lockdown measures on the life satisfaction of the vulnerable group of refugees. This finding brings new insights for the question of vulnerability and resilience of refugees, showing in particular that vulnerable groups may need more support than less vulnerable groups. Several robustness checks have lent credibility to our result. Heterogeneity analyses highlight that gender and support system affect life satisfaction in different ways. Thus, male refugees are impacted more severely than female refugees. Furthermore, we can show that a well-established support system operates as a resilience-enhancing factor. Further research is needed to explore the underlying mechanisms explaining these heterogeneities. Despite the limitations described in Section 4.3, this study provides sound evidence for the vulnerability of refugees during the pandemic and should prompt future studies to examine possible vulnerabilities of further groups to enable adequate policy recommendations to alleviate challenges faced by these vulnerable groups.

## Figures and Tables

**Figure 1 ijerph-19-07409-f001:**
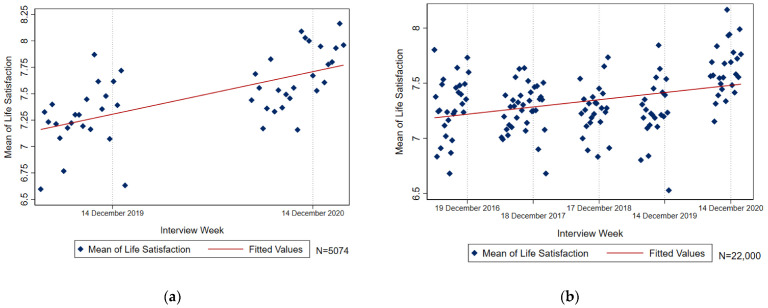
(**a**) Mean of life satisfaction in 2019 and 2020 by interview week, consistent panel; (**b**) mean of life satisfaction from 2016 to 2020 by interview week, all observations.

**Figure 2 ijerph-19-07409-f002:**
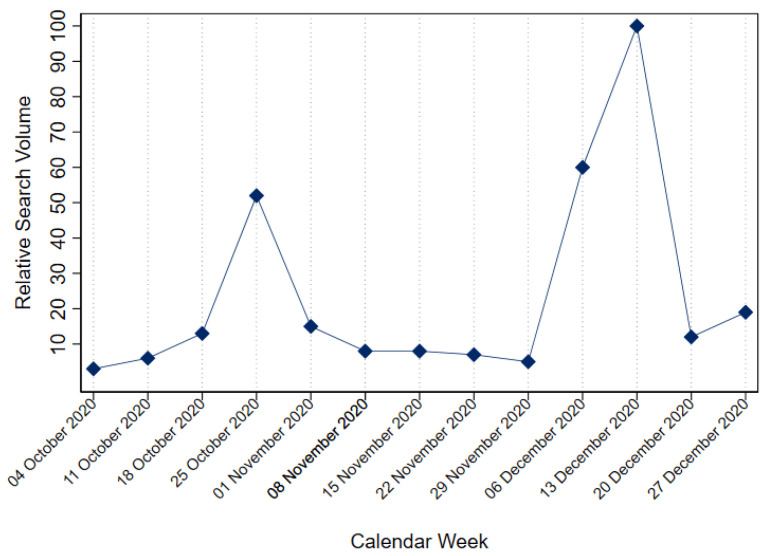
Relative Google-search volume for “Lockdown” in Germany by week.

**Figure 3 ijerph-19-07409-f003:**
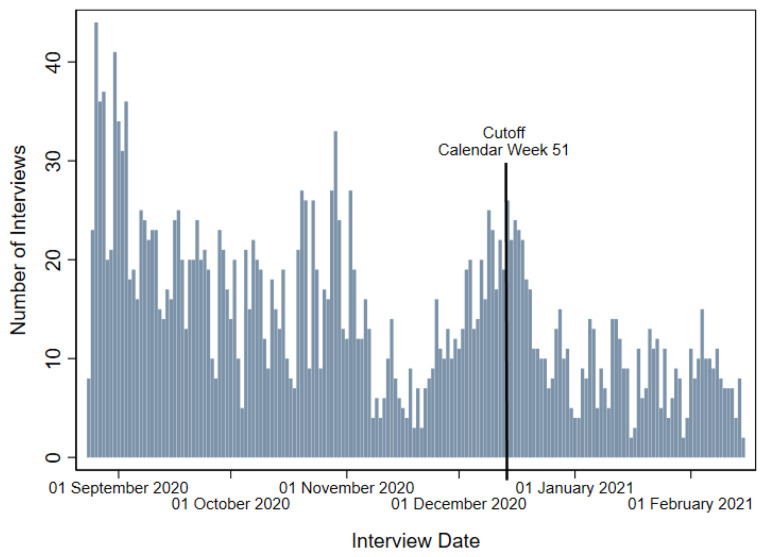
Interviews of the IAB-BAMF-SOEP survey wave 5 conducted in 2020.

**Figure 4 ijerph-19-07409-f004:**
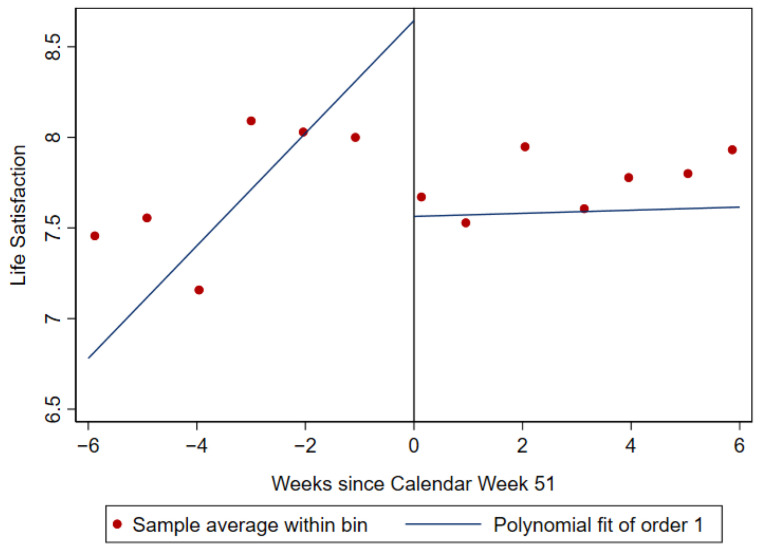
RD plot.

**Figure 5 ijerph-19-07409-f005:**
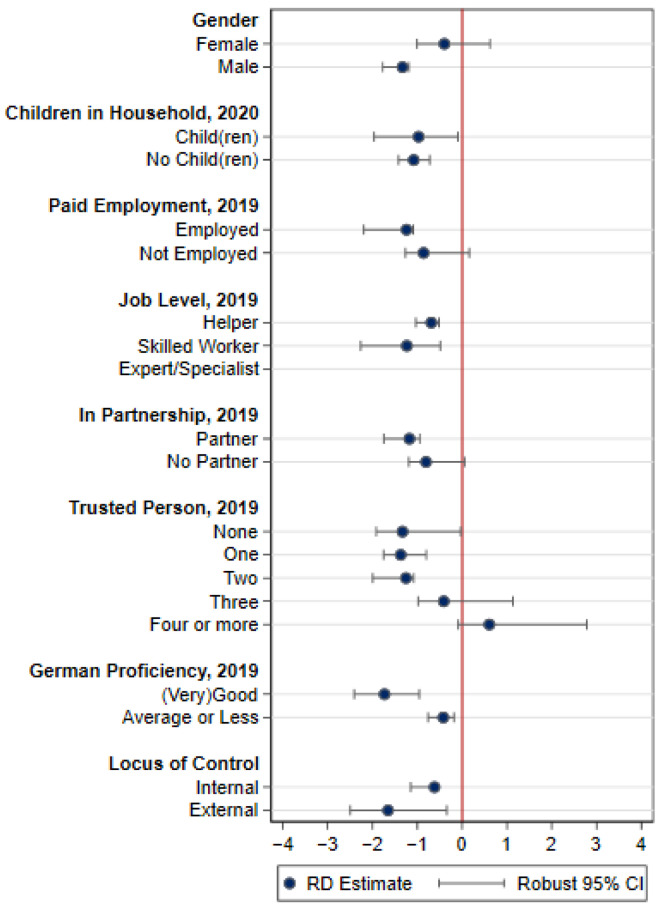
RD estimates by subgroups (Since for refugees employed as experts/specialists the number of observations was too limited to appropriately conduct the RD analysis, we excluded the results for this subgroup from the graphic).

**Table 1 ijerph-19-07409-t001:** Balancedness check for pre-treatment characteristics.

	(1) Mean Control Group	(2) Mean Treatment Group	(3) Difference	(4) ${\hat{ß}}_{RD}$ (ITT)
Female	0.31	0.29	−0.02	0.082
	(0.463)	(0.453)		
Age at Arrival in Years	29.57	29.14	−0.43	−0.354
	(10.451)	(9.536)		
Years of Schooling before Migration	9.05	8.33	−0.72 ***	−0.242
	(4.046)	(4.404)		
In Partnership (2019)	0.6	0.7	0.1 ***	0.072
	(0.49)	(0.459)		
Children under 18 in Household (2019)	0.43	0.5	0.07 **	−0.081
	(0.495)	(0.5)		
German Proficiency, Scale 0–12 (2019)	7.41	7.34	−0.07	−1.362
	(2.866)	(2.698)		
Life Satisfaction (2019)	7.05	7.24	0.19 *	−0.322
	(2.041)	(1.839)		

*Notes*: *** *p* < 0.001, ** *p* < 0.01, * *p* < 0.5. Columns (1) and (2) display mean and standard deviation in brackets for individuals interviewed before and after the lockdown. Column (3) equals the difference between (1) and (2) and displays the result of a *t*-test. Column (4) displays the results of an RD regression using the respective variable as the outcome and interview week as the running variable with week 51 as the cutoff. RD regression is performed with a polynomial of order 1, MSE-optimal bandwidth, and uniform kernel weights. *Data source:* IAB-BAMF-SOEP Survey of Refugees, 2020, weighted.

**Table 2 ijerph-19-07409-t002:** RD effect of lockdown on life satisfaction.

	(1) No Covariates Uniform	(2) No Covariates Triangular	(3) Covariates Uniform	(4) Covariates Triangular
${\hat{ß}}_{RD}$ (ITT)	−0.986	−1.094	−0.757	−0.852
(${\hat{ß}}_{RD}$ in SD)	(−0.551)	(−0.611)	(−0.423)	(−0.476)
95% Robust CI	[−1.242; −0.338]	[−1.483; −0.695]	[−1.093; −0.239]	[−1.208; −0.538]
Robust *p*-Value	0.001	0.000	0.002	0.000
MSE-Optimal Bandwidth	3.283	4.111	3.544	4.119
Eff. Number of Observations	662	763	592	678

*Notes*: Running variable is the calendar week of the interview; outcome is life satisfaction on a scale from 0 to 10. Estimate is the intention-to-treat effect at the cutoff (calendar week 51) estimated with local linear regression with MSE-optimal bandwidth and uniform kernel weights in columns (1) and (3) and triangular kernel weights in columns (2) and (4). Covariates included in columns (3) and (4) are gender, age at arrival, years of schooling before migration, partnership in 2019, children in household in 2019, language proficiency in 2019, and life satisfaction in 2019. The effects in standard deviations were obtained by conducting RD estimations with standardized life satisfaction in wave 2020 (mean 0, std.dev.1) as dependent variable. *Data source:* IAB-BAMF-SOEP Survey of Refugees, 2020, weighted.

**Table 3 ijerph-19-07409-t003:** RD effect with placebo cutoffs.

	(1) Week 40 (28-Sep-2020)	(2) Week 47 (16-Nov-2020)	(3) Week 53 (28-Dec-2020)	(4) Week 02 (11-Jan-2021)
${\hat{ß}}_{RD}$ (ITT)	0.227	0.290	0.201	0.401
95% Robust CI	[−1.305; 1.706]	[−0.376; 0.791]	[−0.227; 1.049]	[−0.294; 0.755]
Robust *p*-Value	0.794	0.485	0.207	0.389
MSE-Optimal Bandwidth	3.648	3.891	3.347	4.719
Eff. Number of Observations	739	589	543	558

*Notes*: Running variable is the calendar week of the interview; outcome is life satisfaction on a scale from 0 to 10. Estimate is the intention-to-treat effect at the respective cutoff estimated with local linear regression with MSE-optimal bandwidth and triangular kernel weights. Covariates included are gender, age at arrival, years of schooling before migration, partnership in 2019, children in household in 2019, language proficiency in 2019, and life satisfaction in 2019. *Data source:* IAB-BAMF-SOEP Survey of Refugees, 2020, weighted.

**Table 4 ijerph-19-07409-t004:** RD design combined with difference-in-difference.

	(1) ±5 Weeks	(2) ±4 Weeks	(3) ±3 Weeks	(4) ±2 Weeks
After Week 51 × Year 2020 (ITT)	−0.837 ***	−1.175 ***	−1.498 ***	−1.735 ***
	(0.217)	(0.222)	(0.234)	(0.26)
After Week 51	0.588 ***	0.741 ***	0.949 ***	1.165 ***
	(0.165)	(0.167)	(0.175)	(0.191)
Year 2020	0.861 ***	1.196 ***	1.477 ***	1.669 ***
	(0.133)	0.142)	(0.156)	(0.181)
Constant	6.969	6.817	6.609	6.512
Number of Observations	1500	1304	1104	854
Adjusted R-Squared	0.029	0.053	0.075	0.094

*Notes:* *** *p* < 0.001. Linear regression with 2019 and 2020 observations where life satisfaction is regressed on a dummy for being interviewed after week 51 (in the respective year), a dummy for the year 2020, and an interaction term of those two. The specifications of the columns differ in their used bandwidth, ranging from five weeks before and after the lockdown in column (1) to two weeks before and after the lockdown in column (4). *Data source:* IAB-BAMF-SOEP Survey of Refugees, 2019–2020, weighted.

**Table 5 ijerph-19-07409-t005:** RD effect with placebo outcome variable interview mode.

	Outcome: Interview Mode
${\hat{ß}}_{RD}$ (ITT)	−0.020
95% Robust CI	[−0.169; 0.195]
Robust *p*-Value	0.887
MSE-Optimal Bandwidth	3420
Eff. Number of Observations	662

*Notes*: Running variable is the calendar week of the interview; outcome is interview mode (0 = CAPI, 1 = CAPI-TEL). Estimate is the intention-to-treat effect at cutoff week 51 estimated with local linear regression with MSE-optimal bandwidth and triangular kernel weights. *Data source:* IAB-BAMF-SOEP Survey of Refugees, 2020, weighted.

## Data Availability

This study uses the factually anonymous data of waves 2016–2020 of the IAB-BAMF-SOEP Survey of Refugees. The survey is conducted jointly by the Institute for Employment Research (IAB), the research data center of the Federal German Office for Migration and Refugees (BAMF), and the German Socio-Economic Panel (SOEP) at the German Institute for Economic Research (DIW). External researchers may apply for access to these data by submitting a user-contract application to the Research Data Centre (FDZ) of the German Federal Employment Agency (BA) at the Institute for Employment Research (IAB). DOI: 10.5684/soep.iab-bamf-soep-mig.2020.

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
