# Peer review of "Resilient or Vulnerable? Effects of the COVID-19 Crisis on the Mental Health of Refugees in Germany"

_ijerph, 2022, doi:10.3390/ijerph19127409_

Round 1

Reviewer 1 Report

26/27 missing citation

41 biased statement – It can be argued there is no need for universal definition anyway!

48 what is the correlation between resilience and mental functioning

50 what do you mean you understand resilience, resilience is multifaceted and can only be defined through a contextual perspective.

52 second section instead of second step

67 What type of refugees, political, geopolitical or natural disaster refugees or economical refugees.

Introduction should clearly define the research problem and argument which is to a certain extend was missing! The importance

93 what is the paper trying to measure? The consequences of the government procedures on mental health? Not very clear? Measuring the COVID19 pandemic in general is difficult task?

97 – 99 rewrite to articulate the argument better

The variables of the survey ( IAB-BAMF-SOEP) are dependent and not designed to measure resilience?

139 Life satisfaction is already impacted, how can you differentiate this variable from before migrating and after migrating.

143 asked in general not in relation to the pandemic

187 google search is not an indication of identifying the majority trend behavior!

194 what do you mean by well notice, how did this come about?

327 how the paper will bridge the gap between quan to qual?

345 provide further evidence or explain more

467 Lockdown has a negative effect on everybody’s life 

Reviewer 2 Report

This relevant paper describes the method well and presents interesting results. However, the first part of the paper in particular lacks an appropriate embedding in literature, especially an explanation and embedding of the use of the resilience term. This weakness continues in the discussion and interpretation of the results. In particular, the discussion and policy implications lack an embedding in relevant studies that deal specifically with this target group. This would further support the importance of the paper.

p. 1-2: Possible definitions of resilience are explained, then reference is made to the definition of Bonanno. However, there is no conclusive derivation for the decision in favour of Bonanno in the context of the specific paper and the further procedure (first and second step refer to different approaches to resilience). In particular with regard to the second step, a definition of vulnerability in the specific context is missing.

p.2: Among the list of possible traumatising experiences, the integration process itself is not addressed, although it can represent a central factor in the sense of its concurrence with the lockdown.

p.3 and 4: In connection with the data used, life satisfaction is referred to as a dimension of resilience. However, life satisfaction must be perceived as one dimension among others. Here again, the importance of a more comprehensive description and contextualisation of the concept of resilience becomes clear.

p.5: The usefulness of using google search data to draw conclusions about perception can be questioned or is unnecessary. As mentioned in the text, it is also sufficient to point out the limitations of the work.

Discussion and policy implications: see above.

Round 2

Reviewer 1 Report

I believe that all comments have been addressed by authors